# Parental Psychological Control and Adolescent Academic Achievement: The Mediating Role of Achievement Goal Orientation

**DOI:** 10.3390/bs14030150

**Published:** 2024-02-20

**Authors:** Lingruina Xu, Jinshan He, Xuejiao Wei, Yunyun Zhang, Li Zhang

**Affiliations:** 1School of Society and Psychology, Central University of Finance and Economics, Beijing 100081, China; 2021312011@email.cufe.edu.cn (L.X.); 2022212208@email.cufe.edu.cn (J.H.); 2020212197@email.cufe.edu.cn (X.W.); 2Collaborative Innovation Center of Assessment for Basic Education Quality, Beijing Normal University, Beijing 100875, China

**Keywords:** parental psychological control, adolescents, academic achievement, love withdrawal, guilt induction, authority assertion

## Abstract

This study examined the multiple mediating roles of achievement goal orientation between three parental psychological control (PPC) strategies and adolescents’ academic achievement. The study sample consisted of 2613 Chinese middle school adolescents (52.6% boys) who were followed for one and a half years; they completed questionnaires on PPC (including love withdrawal, guilt induction, and authority assertion), achievement goal orientation (involving the mastery approach, the performance approach, and performance-avoidance goals), and academic achievement. We found that (1) the direct effects of the three strategies on academic performance differed, with love withdrawal directly and negatively predicting adolescents’ academic achievement and guilt induction and authority assertion not being significant direct predictors. (2) The mediating role of achievement goal orientations differed across the psychological control strategies. Specifically, love withdrawal led to adolescents’ academic achievement through their performance-approach goal orientation, performance-avoidance goal orientation, and mastery goal orientation. Moreover, guilt induction and authority assertion had impacts only on adolescents’ performance-approach and performance-avoidance goal orientations. This study highlights the negative impact of love withdrawal on adolescents’ internal motivation and academic achievement by warning parents not to use this strategy to influence their children’s thoughts and feelings.

## 1. Introduction

Substantial evidence suggests that parenting practices, as an essential part of children’s growth process, play a crucial role in children’s academic achievement [1,2,3]. In studies from Western countries, parental psychological control (PPC), a typical negative parenting practice, has been well-documented to have a negative impact on youth academic development [4,5,6]. However, in East Asia, studies on the relationship between PPC and children’s academic achievement are limited, and the results are inconsistent.

Psychological control predicts negative academic performance among elementary school students [7], high school students [8], and college students [9]. However, additional cross-cultural studies indicate no significant correlation between psychological control and academic performance [10,11,12,13]. For example, Wang et al. found that Chinese parents’ psychological control did not significantly predict a decline in academic functioning in a 6-month longitudinal study of first-year middle school students [12]. A similar failure to make a significant prediction was found among high school students [11,13]. A study of Korean elementary school students revealed no direct relationship between psychological control and children’s academic performance [10].

Previous research has compared different psychological control strategies less often to respond to controversy. According to previous studies, we are concerned about the following two questions:(1)Will parental psychological control strategies have an impact on a child’s academic performance in Chinese culture?(2)How does parental psychological control affect students’ academic performance?

For the present research, we divided psychological control into three strategies to examine whether the inconsistency of the relationship was influenced by the type of psychological control. Furthermore, we explored the role of achievement goal orientation in the relationship between different kinds of psychological control and academic performance using three-wave longitudinal data in the Chinese cultural context.

### 1.1. Parental Psychological Control and Academic Achievement

PPC involves attempting to control one’s child through psychological tactics; it is intrusive and manipulative for children’s psychological and emotional worlds, frustrates children’s needs, disrupts their autonomous process, and creates vulnerability to maladjustment [14]. In the West, researchers typically use six aspects (constraining verbal expression, invalidating feelings, personal attacks, guilt induction, love withdrawal, and erratic emotional behavior) to measure psychological control [15]. Considering that different cultural backgrounds in different countries create different family education styles, some researchers have found that guilt induction, love withdrawal, and authority assertion are frequently used psychological control strategies in China [12].

Many Western empirical studies suggest that the PPC predicts poor academic functioning. Children of psychologically controlling parents are more likely to exhibit maladaptive academic behaviors, such as more negative attitudes toward school [16] and poorer grades [4,5]. Psychological control not only exacerbates fear of failure [6] but also leads to learned helplessness due to constant frustration [17], both of which are not conducive to strong academic performance.

According to self-determination theory (SDT), this negative prediction can be well understood. PPC hinders the satisfaction of individuals’ competence, autonomy, and relatedness needs and disrupts the generation of intrinsic motivation [18]. SDT maintains that an understanding of human motivation requires consideration of innate psychological needs for competence, autonomy, and relatedness [19]. Fulfilling these three needs results in enhanced self-motivation and psychological well-being; if these needs are hindered, individuals experience decreased motivation and well-being [20]. Children who suffer from PPC feel that they are forced to act, feel, or think in ways that are dictated by their parents [18]. Therefore, PPC has adverse effects on emotional and academic functioning.

However, in East Asia, studies on the relationship between PPC and children’s academic performance are limited, and the results are inconsistent. Some studies imply that the negative effects of psychological control can be generalized to non-Western cultures [6,7,8,9,21]. Barber et al. [21] showed that psychological control is positively associated with both internalized and externalized problem behaviors in 10 different countries spanning every continent of the globe. Moreover, psychological control predicts negative academic performance among elementary school students [7], high school students [8], and college students [9]. However, in the past decade, the focus of related research has gradually shifted to East Asia, particularly China, and several cross-cultural studies have shown inconsistent results. A few studies have shown a nonsignificant, negative association between PPC and academic achievement [10,11,12,13] and found that the Chinese PPC did not significantly predict poorer academic performance in a six-month longitudinal study comparing first-year middle school students in China and the U.S. Other researchers have reported similar outcomes among high school students [11,13]. Lee et al. [10] reported no direct relationship between psychological control and children’s academic performance in a study of Korean elementary school students.

The inconsistent findings in the literature may stem in part from cultural differences. In collectivist cultures, strategies that emphasize parental control and authority are consistent with universal socialization goals and may, therefore, be less maladaptive [22]. The negative effects of this parenting style can be mitigated when adolescents understand parental control as an expression of care and reasonable concern.

However, previous research has less often compared different strategies of psychological control, which is another important reason for the inconsistent findings in the literature. First, Eastern and Western parents do not use the same psychological control tactics, with Chinese parents employing more love withdrawal strategies [23]. Ng et al. found that Chinese parents use more control than American parents do, in part because Chinese parents’ sense of value is more dependent on their children’s performance [24]. Second, the negative outcomes of different types of psychological control are not equivalent, and their impact on academic performance varies across cultures. For example, Jiang and Lee explored two psychological control strategies commonly used in Korean culture: autonomous and emotional control [25]. They found that the two types of control have different effects, with emotional control being significantly and positively associated with children’s feelings of helplessness and autonomous control significantly reducing academic self-efficacy. Similarly, Fang and Lau scrutinized two forms of PPC: hostile psychological control (criticism, interference, invalidation) and relational induction (guilt induction, shaming, reciprocity, social comparison) [22]. Hostile psychological control is associated with behavioral problems only in European-American families; relational induction is not significantly associated with problematic conduct in Hong Kong (HK) Chinese and European-American families.

Influenced by Confucian values, Chinese parents rely more on authority assertion and love withdrawal [26]. In addition, we addressed guilt induction as a representation of classical psychological control strategies in the present research. Hence, our first aim was to investigate the effects of these three psychological control strategies on academic achievement in a Chinese sample.

### 1.2. Achievement Goals as Mediators

How does PCC relate to academic achievement? One possible explanation is that achievement goals play a mediating role. Early researchers divided achievement goal orientation into mastery goal orientation—which focuses on developing new skills, learning new knowledge, and improving one’s abilities—and performance goal orientation—which centers on external evaluations, whether one is better than others, and outcomes [27]. Furthermore, Elliot and Harackiewicz divided performance goal orientation into two dimensions, the performance-approach orientation and the performance-avoidance orientation, whereby the performance-approach orientation is utilized to make one better than others, while the latter seeks to avoid being worse than others [28].

There are two reasons to support our view. First, psychological control is associated with achievement goal orientation. For example, research indicates that parental control and authoritarian parenting styles may lead adolescents to exhibit more performance-approach and performance-avoidance orientations [29,30]. In addition, college students’ fear of failure is associated with the withdrawal of perceived parental love, and this fear of failure leads to an avoidance orientation [6,31]. Another study in the field of sports revealed that psychological control is related not only to performance-approach and performance-avoidance goals but also to mastery goal orientation [32].

Second, achievement goal orientation is related to academic achievement [33,34]. Different achievement goals predict different academic outcomes. Specifically, mastery goal orientation helps individuals perform well academically, while performance-avoidance orientation negatively predicts academic achievement [31,35,36]. The results of studies on the relationship between performance-approach orientation and academic achievement are inconsistent. Several studies have shown that self-compassion is associated with positive academic performance [37]. Instead, it is considered to be related to test anxiety, shallow learning processes, and low academic performance [38]. In response to this controversy, Senko and Tropiano proposed the goal complex model, which applies SDT and considers the reasons for pursuing a goal [39]. When individuals outperform others for controlling reasons (e.g., rewards, proving oneself, impressing others), this can result in maladaptive consequences (e.g., help avoidance). However, outperforming others for autonomous reasons (e.g., fun, a challenge, personal usefulness) can lead to adaptive outcomes (e.g., self-efficacy). According to this theory, under PPC, adolescents pursue performance goals more for controlling reasons.

In recent years, several studies have explored the mediating role of achievement goal orientation. Xiang et al. [40] found a nonsignificant mediating effect of achievement goal orientation in which performance-approach goals acted as suppressors between PPC and certain school adjustment variables [40]. However, Xu et al. [41] obtained a different outcome: PPC led to adolescents’ maladaptive academic functioning, mainly through their performance-approach and performance-avoidance goal orientations rather than mastery goal orientation. In view of the shortcomings and inconsistencies of the above research and because few studies have examined the mediating mechanism of each strategy separately, our second purpose is to examine the mediating role of achievement goal orientation between parental psychological control and academic achievement.

### 1.3. The Present Study

In sum, findings on the effects of psychological control on adolescents’ academic performance are inconsistent. Moreover, previous studies have compared different psychological control strategies less often, and it is unclear whether the three dimensions of guilt induction, authority assertion, and love withdrawal all negatively predict academic performance in Chinese culture. Additionally, studies that have investigated their mediating role on this basis are more limited, especially in-depth studies on the dimensions of psychological control; specific mechanisms are not yet apparent. Thus, we conducted a longitudinal study of one-and-a-half-year periods with a sample of first-year Chinese adolescents because the impact of PPC on academic achievement is crucial as individuals in the first year enter early adolescence and are in the transition stage from elementary to secondary school. Clarifying the key factors that influence academic underachievement and their mechanisms of action will help to develop targeted preventive interventions and related policies to reduce children’s academic underachievement and boost academic performance.

Based on existing research, we formulated the following hypotheses:(1)The three psychological control strategies (guilt induction, authority assertion, and love withdrawal) differ in their direct predictions of academic achievement.(2)Achievement goal orientation mediates the relationship between different psychological control strategies and academic achievement through different pathways of action for performance-approach, performance-avoidance, and mastery goal orientations.

## 2. Materials and Methods

### 2.1. Participants

We recruited participants from 47 Grade One classes at 7 middle schools (2 urban schools, 1 county school, and 4 township schools) from two districts and counties in a province in central China. After effective screening, 2613 students participated at T1, and the missing data were filled in by maximum likelihood estimation. Among the students, 1375 (52.6%) were male and 1238 (47.4%) were female. There were 2166 (82.9%) families with agricultural household registration, 378 (14.5%) families with nonagricultural household registration, and 69 (2.6%) missing data. There were 462 (17.7%) only children, 2134 (81.7%) non-only children, and 16 (0.6%) missing data. For the level of parental education, 368 fathers (14.1%) had a primary school education or below, 1526 (58.4%) had a junior middle school education, 561 (21.4%) had a senior high school or secondary school degree, 142 (5.5%) had a college degree or above, and 16 (0.6%) were missing. In addition, 482 mothers (18.5%) had a primary school education or below, 1472 (56.3%) had a junior middle school education, 470 (18.0%) had a senior high school education or secondary school degree, 32 (1.2%) had a college degree or above, and 23 (0.9%) were missing.

### 2.2. Measurement Instruments

Our data were collected at three different time points. We administered the second survey in the first semester (T1), the first semester of the junior year (T2), and the second semester of the junior year (T3).

#### 2.2.1. Parental Psychological Control

We assessed PPC at T1 using the questionnaire adopted by Wang et al. [12], which has good reliability and validity among Chinese students [24]. We measured PPC with 18 items, 10 of which gauged guilt induction (e.g., “When I didn’t live up to my parents’ expectations, they told me that I should feel guilty”), 5 of which established love withdrawal (e.g., “If I did something my parents didn’t like, they would appear cold and unfriendly”), and 3 of which determined authority assertion (e.g., “My parents told me they wanted me to do what was best for me and that I shouldn’t question it”). In this study, the Cronbach’s alpha for PPC was 0.90.

#### 2.2.2. Achievement Goal Orientation

We assessed achievement goal orientation at T2 using a revised version of the Patterns of Adaptive Learning Survey (PALS) [42]. The Patterns of Adaptive Learning Scales (PALS) were first published in 1997 [43], and in 2000, the dimension of personal achievement goals was revised to optimize the number of items in the scale and improve its internal consistency [42]. Our study used the version of PALS in 2000, which includes 3 dimensions and 14 items, namely mastery goals (5 items, such as “It is important for me to fully understand my work”), performance approaches (5 items, such as “one of my goals is to show my classmates that I am good at my work”), and performance avoidance (4 items, such as “It is important for me not to look stupid in class”). The items were rated on a 5-point scale from 1 (totally inconsistent) to 5 (totally consistent), with higher scores reflecting higher levels. In the current sample, the Cronbach’s alphas of each aspect of achievement goal orientation were 0.91 and 0. 90, and 0.70, respectively.

#### 2.2.3. Academic Achievement

Academic achievement involves one’s final grades for the second semester of the junior year provided by the school, including Chinese, mathematics, English, history, biology, geography, and politics. The final exams of all schools are often unified into a single test with the same test paper content and answer times so that there is strong comparability between the outcomes. We standardized and added the results of the seven subjects as the academic performance of each student.

### 2.3. Procedure and Data Analysis

We evaluated the participants in a group by employing unified instructions. Due to variations in the timing of questionnaire collection across different schools and classes, we have established a uniform time frame of one week to ensure relative consistency across the various time points during the three data-tracking sessions. After the testers briefly explain the purpose and significance of the survey to the participants, all participants are required to complete the questionnaire within the duration of one class period. In the first semester of junior high school, students completed the PPC scale, and in the first semester of their junior year, students completed the achievement goal orientation scale. Half a year later, we collected students’ final scores. We conducted descriptive statistics and correlation analysis via SPSS 22.0. We used Amos 21.0 for mediation analysis to test the mediating role of achievement goal orientation between psychological control and academic achievement. In this study, we utilized a sample of 2613 middle school students, with the number of missing values for each variable ranging between 16 and 69, representing a missing percentage of 0.6% to 2.6%. Since this meets the requirement for the application of the maximum likelihood estimation method, which states that the number of missing values for a single variable should not exceed 5%, our use of the maximum likelihood estimation method is justified.

## 3. Results

### 3.1. The Common Method Deviation Test

Since we obtained the psychological control and achievement goal orientation of the participants through self-reports, we used the Harman single-factor test and confirmatory factor analysis with common method factors to test common method bias. We obtained nine factors with characteristic roots greater than 1 without rotation, which explained 63.53% of the variance in total. The variance explained by the first factor was 17.72% (<40%). This outcome indicates that there was no serious common method bias in this study.

### 3.2. Descriptive Statistics

We utilized descriptive statistics and correlation analyses for each variable; the results are presented in Table 1. PPC was significantly and positively correlated with performance-approach and performance-avoidance orientations and negatively correlated with mastery goal orientation and academic achievement. Academic achievement was significantly and positively correlated with mastery goal orientation and negatively correlated with performance-approach orientation, performance-avoidance orientation, and PPC. Gender and family economic status were significantly correlated with academic performance. In addition, we use gender as a moderating variable to further explore whether gender has a systematic impact on the results, but the results are not significant, indicating that gender does not have a systematic impact. Therefore, we control for the above variables in the subsequent analysis.

### 3.3. Direct Effect Test

After controlling for the influence of gender and family economic status, we examined the direct effect of PPC on academic performance. The results (Table 2) imply that only the love withdrawal dimension was significantly negatively related to youth academic performance (β = −0.14, *p* < 0.001), while guilt induction and authority assertion did not predict academic achievement (β = −0.03, *p* = 0.28, β = −0.04, *p* = 0.12).

### 3.4. Longitudinal Mediating Effect Analysis

Our second goal was to explore the inner mechanisms of psychological control and academic performance. We constructed three models to investigate the mediating roles of T1 psychological control and T3 academic achievement when controlling for the effect of gender and family economic status on T3 academic performance. We used the three dimensions of T1 psychological control as predictor variables; T2 performance-approach orientation, mastery goal orientation, and performance-approach orientation as mediating variables; and T3 academic achievement as the dependent variable.

We initially tested the measurement models; they provided an acceptable fit to the data (when T1 guilt induction was the prediction variable: χ^2^/df = 3.42, *p* < 0.01; TLI = 0.98, CFI = 0.99; RMSEA = 0.03; when T1 authority assertion was the prediction variable: χ^2^/df = 3.63, *p* < 0.01; TLI = 0.97, CFI = 0.99; RMSEA = 0.03; and when T1 love withdrawal was the prediction variable: χ^2^/df = 2.94, *p* < 0.01; TLI = 0.98; CFI = 0.99; RMSEA = 0.03).

Figure 1 outlines when T1 guilt induction was the prediction variable, the paths for this model, and the measurement and standardized path coefficients. Achievement goal orientation played a partial intermediary role between guilt and academic performance; the mediating effect was −0.05 (95% CI: −0.06, −0.03). The mediating effects of performance-approach and performance-avoidance goal orientations were −0.03 (95% CI: −0.04, −0.02) and −0.01 (95% CI: −0.02, −0.003), respectively. The mediating effect of mastery goal orientation was not significant, and the mediating effect of guilt induction on academic performance accounted for 0.05/0.16 = 31.25% of the total effect.

Figure 2 shows that when T1 love withdrawal was the prediction variable, the paths were included in this model, and the measurements and standardized path coefficients were calculated. Achievement goal orientation played a partially intermediary role. The mediation effect was −0.09 (95% CI: −0.11, −0.07). For the performance-approach goal, it was −0.02 (95% CI: −0.04, −0.02). The mastery goal orientation was −0.06 (95% CI: −0.07, −0.04). The performance-avoidance goal orientation was −0.01 (95% CI: −0.02, −0.003). The mediating effect of guilt induction on academic performance accounted for 0.09/0.18 = 50% of the total effect.

Figure 3 illustrates that T1 authority assertion was the prediction variable, the paths for this model, and the measurement and standardized path coefficients. Achievement goal orientation plays a partial intermediary role between authority assertion and academic performance. The mediation effect was −0.02 (95% CI: −0.04, −0.002). The mediating effects of performance-approach and performance-avoidance goal orientations were −0.02 (95% CI: −0.03, −0.01) and −0.01 (95% CI: −0.02, −0.003), respectively. The mediating effect of mastery goal orientation was not significant. The mediating effect of authority assertion on academic performance accounted for 0.02/0.13 = 5.38% of the total effect.

## 4. Discussion

For this study, we conceptualized PPC as a multidimensional construct (i.e., guilt induction, authority assertion, and love withdrawal), and we examined the associations of each of these elements with adolescents’ achievement in junior high school. This study contributes to the literature by identifying which types of PPC are most harmful to junior high school students and the mechanisms through which these types of PPC operate. We found that the direct effects of the three strategies on academic achievement differed, with love withdrawal directly and negatively predicting adolescents’ academic achievement, whereas guilt induction and authority assertion were not significant direct predictors. Furthermore, the mediating role of achievement goal orientations differed across psychological control strategies. Specifically, love withdrawal led to adolescents’ academic performance through their performance-approach and performance-avoidance goal orientations, and mastery goal orientation. Moreover, guilt induction and authority assertion had effects only on adolescents’ performance-approach and performance-avoidance goal orientations.

### 4.1. Parental Psychological Control and Academic Achievement

We found that love withdrawal in PPC can directly and significantly negatively predict adolescent academic achievement, which is consistent with the findings of previous studies [8,9]. Guilt induction and authority assertion did not predict academic performance, and many studies have reported similar results [10,11,12,13]. In general, the three psychological control strategies have impacts on academic achievement with different predictive effects, which is consistent with the first hypothesis of this study. Furthermore, we discovered different predictive outcomes for different types of psychological control, which partially explains the differences in the findings of prior research. Many scholars claim that autonomy is not highly valued in the East; therefore, PPC might not be as detrimental to children’s academic functioning as found in Western samples [23]. The negative effects of this parenting style can be mitigated when Chinese adolescents understand psychological control as an expression of care and legitimate concern. Guilt induction and authority assertion are more easily understood as such concerns than love withdrawal and are therefore not as harmful. Love withdrawal is significantly and positively associated with children’s feelings of helplessness [44], and for children, parents convey conditional love to their children, which is not conducive to a secure and stable attachment relationship. Moreover, this insecurity can have a negative impact on the child.

### 4.2. The Mediating Effects of Students’ Achievement Goal Orientations

Our findings support the mediating role of achievement goal orientation between psychological control and academic achievement, in which guilt induction and authority assertion influence academic achievement through adolescents’ performance-approach and performance-avoidance goal orientations. This finding is consistent with the results of previous studies [41]. However, inconsistent with the outcomes of Xu et al. [41], the psychological control strategy of love withdrawal affects adolescents’ academic performance not only through adolescents’ performance-approach and performance-avoidance goal orientations but also through mastery goal orientation, where we observe a significant and negative effect of love withdrawal on mastery goal orientation. In summary, achievement goal orientation mediates the relationship between different psychological control strategies and academic achievement through distinct pathways of action for performance-approach, performance-avoidance, and mastery goal orientations, which is consistent with the second hypothesis of this study.

This finding can be understood in light of attachment theory. Elliot and Reis applied Bowlby’s theory to adulthood, arguing that attachment relationships influence adult motivation to meet competence needs [45]. Specifically, these theorists maintain that secure attachment promotes motivation to pursue competence, a need for achievement in the face of tasks and a goal of mastery, and a willingness to work to attain goals. On the other hand, insecure attachment redirects competence motivation to a self-protective concern for avoiding incompetence, manifested in fear of failure and performance-avoidance goals. Parental love withdrawal strategies impair secure attachment, difficulty developing mastery orientation, and a lack of desire to acquire knowledge or improve skills. Furthermore, empirical studies indicate that college students’ fear of failure is associated with the perceived withdrawal of parental love [31]; this fear of failure leads to low self-efficacy, making students afraid and unconvinced of their ability to master a piece of knowledge or skill.

Moreover, all three control strategies are associated with a performance motivation orientation because parents who control their children tend to care more about their children’s grades than about their skills and tend to overpunish or praise their children to encourage them to excel academically [46]. Consequently, adolescents believe that learning is about meeting parental expectations, avoiding harsh punishments, and simply receiving good grades.

### 4.3. Limitations and Future Research

This study has several limitations. First, we did not distinguish which of the psychological controls of fathers and mothers would have a different impact on children’s achievement. Previous studies have shown that due to differences in parental values, Chinese parents tend to exert greater psychological control over their children’s academic performance compared to American parents [26]. Furthermore, the effects of paternal and maternal psychological control on children’s academic functioning may be interactive [7]. Therefore, future research could explore how paternal and maternal psychological control may have different effects on students’ academic achievement. Second, we obtained all the data from a single source, which may influence the objectivity of our findings. Although some studies suggest that data on psychological control are best reported by adolescents on their own, future studies should use multiple sources of assessment, such as observational data, to avoid common method bias. Finally, most of the participants were students in rural areas, and the results have yet to be validated in additional studies in urban areas.

### 4.4. Implications

Despite these limitations, our study has important practical and theoretical implications. First, it provides empirical validation and an extension of SDT and achievement goal orientation theory. Second, individuals in the first grade enter early adolescence and are in the transition phase from elementary to secondary school, where PPC is critical for academic performance. We emphasize the dual negative effects of love withdrawal strategies through PPC on adolescents’ internal motivation and academic achievement. This study warns parents against using love withdrawal strategies to influence their children’s thoughts and feelings.

## 5. Conclusions

In this study, we conceptualized PPC as a multidimensional construct (i.e., guilt induction, authority assertion, and love withdrawal), and we examined the associations of each of these elements with adolescents’ achievement in junior high school. This study contributes to the literature by identifying which types of PPC are most harmful to junior high school students and the mechanisms through which these types of PPC operate. We found that the direct effects of the three strategies on academic achievement differed, with love withdrawal directly and negatively predicting adolescents’ academic achievement, whereas guilt induction and authority assertion were not significant direct predictors. Furthermore, the mediating role of achievement goal orientations differed across psychological control strategies. Specifically, love withdrawal led to adolescents’ academic performance through their performance-approach and performance-avoidance goal orientations, and mastery goal orientation. Moreover, guilt induction and authority assertion had effects only on adolescents’ performance-approach and performance-avoidance goal orientations.

## Figures and Tables

**Figure 1 behavsci-14-00150-f001:**
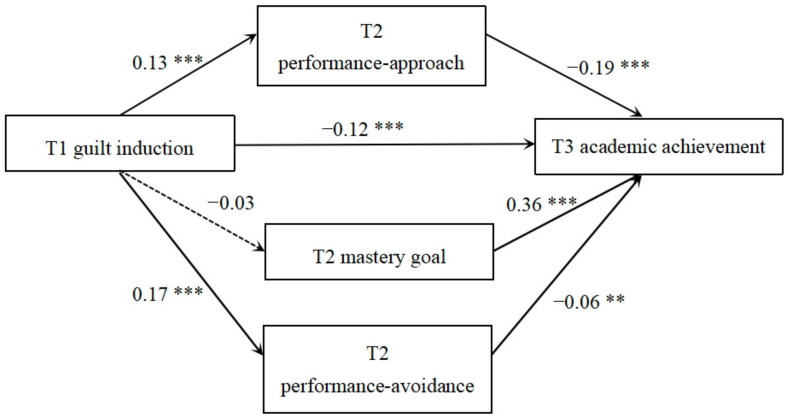
Longitudinal mediating model of guilt induction and academic achievement. Note: ** *p* < 0.01, *** *p* < 0.001.

**Figure 2 behavsci-14-00150-f002:**
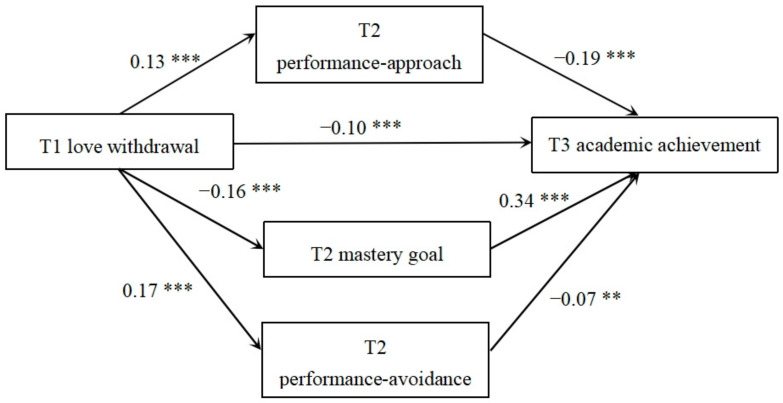
Longitudinal mediating model of love withdrawal and academic achievement. Note: ** *p* < 0.01, *** *p* < 0.001.

**Figure 3 behavsci-14-00150-f003:**
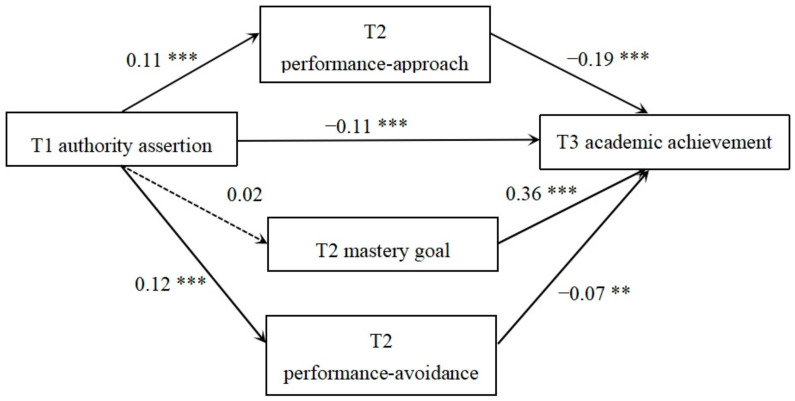
Longitudinal mediating model of authority assertion and academic achievement. Note: ** *p* < 0.01, *** *p* < 0.001.

**Table 1 behavsci-14-00150-t001:** Descriptive statistics and correlation analyses for the observed variables.

	1	2	3	4	5	6	7	8	9	10
1. Gender	1									
2. Family economic status	0.002	1								
3. T1 psychological control	−0.111 **	0.008	1							
4. T1 guilt induction	−0.117 **	0.011	0.953 **	1						
5. T1 love withdrawal	−0.096 **	−0.004	0.844 **	0.686 **	1					
6. T1 authority assertion	−0.050 *	0.013	0.761 **	0.667 **	0.487 **	1				
7. T2 performance approach	−0.212 **	−0.061 **	0.143 **	0.134 **	0.128 **	0.105 **	1			
8. T2 mastery goal	0.071 **	0.008	−0.067 **	−0.027	−0.162 **	0.024	0.104 **	1		
9. T2 performance avoidance	−0.181 **	−0.036	0.180 **	0.170 **	0.169 **	0.116 **	0.601 **	−0.015	1	
10. T3 academic achievement	0.203 **	0.044 *	−0.197 **	−0.174 **	−0.197 **	−0.137 **	−0.228 **	0.346 **	−0.220 **	1

Note: * *p* < 0.05, ** *p* < 0.01.

**Table 2 behavsci-14-00150-t002:** Regression of psychological control and academic performance.

	Model 1		Model 2	
	β	*t*	β	*t*
Constant		−7.94 ***		−1.25
Gender	0.20	10.59 ***	0.18	9.694 ***
Family economic status	0.04	2.27 *	0.04	2.33 *
Independent variables				
Guilt induction			−0.03	−1.09
Love withdrawal			−0.14	−5.28 ***
Authority assertion			−0.04	−1.55
R squared	0.04		0.08	
F	58.70 ***		43.67 ***	
Adjusted R squared	0.04 ***		0.08 **	

Note: * *p* < 0.05, ** *p* < 0.01, *** *p* < 0.001.

## Data Availability

The data that support the findings of this study are contained within the article and are available from the corresponding author upon reasonable request.

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
