# Peer review of "Parental Psychological Control and Adolescent Academic Achievement: The Mediating Role of Achievement Goal Orientation"

_behavsci, 2024, doi:10.3390/bs14030150_

Round 1

Reviewer 1 Report

Comments and Suggestions for Authors

The authors have done a very nice job in describing the need and  rationale for the study.  The relevance of study findings from the West  to East Asia is important for this research as well as bridging the gap that often exists in social sciences as far as relevance of research across cultures. Certainly the findings of this study broaden our understanding of  overall parenting practices.

The authors need to clarify a couple of things stated in the manuscript.

1. Line 208: Please elaborate what " We assessed PPC at Wave 1 using Wang et al. (2007)" means.

2. Line 217: Please clarify revised version of the PALS. What is the original version of the PALS, and is what is presented  in lines 217-225, the revised version? If so, what is different in the revision in comparison to the original? There does not appear to be authorship attributed to the PALS, so indicating that would be important.

3. Line 194: Missing data were filled in by maximum likelihood estimation (MLE). What percentage of the data were missing that required use of MLE? It would be important to indicate the percentage of data for each key variable that were missing and required MLE.

4. line 234-235- what was the rationale for having participants completing the questionnaire within a specified time frame I(what was the time frame?) and what impact did that have on the data collected and responses provided?

5. Line 399-A statement here about which parent typically exerts more  psychological control in Chinese culture (if known)  and/or more likely to direct their child's academic achievement would contextualize the importance of this limitation identified by the authors.

Reviewer 2 Report

Comments and Suggestions for Authors

Dear author(s),

I have read with much interest your paper titled “Parental psychological control and adolescent academic achievement: The mediating role of achievement goal orientation.

The paper present useful data regarding the effects of the three psychological control strategies on academic achievement in a Chinese sample and the mediating role of achievement goal orientation between parental psychological control and academic achievement.

However, there are some minor points that I noticed should be improved: 

1.     Line 154: Since the hypotheses are mentioned in line 181-186, I recommend removing “We therefore hypothesized that a performance-approach goal would be negatively related to academic achievement” from line 154.

2.     Line 192: I suggest moving the paragraph “We administered the second survey in the first semester (T1), the first semester of the junior year (T2), and the second semester of the junior year (T3).” to the section “2.2. Measurements instruments”.

3.     It would also be useful to mention the research questions in relation to the research purpose and  hypotheses.

4.     Also, if you agree, it would be useful to specifically mention in the Conclusions section the confirmation/refirmation of the research hypotheses by the research results.
